# Person Re-Identification via Pyramid Multipart Features and Multi-Attention Framework

**Randa Mohamed Bayoumi** [1,*] , **Elsayed E. Hemayed** [2,3] , **Mohammad Ehab Ragab** [1] **and Magda B. Fayek** [2]

1 Informatics Research Department, Electronics Research Institute, Giza 12622, Egypt; mehab@hotmail.com
2 Computer Engineering Department, Faculty of Engineering, Cairo University, Giza 12613, Egypt; hemayed@ieee.org (E.E.H.); magdafayek@ieee.org (M.B.F.)
3 Zewail City of Science and Technology, University of Science and Technology, Giza 12578, Egypt
* Correspondence: rm2520@aucegypt.edu

**Abstract:** Video-based person re-identification has become quite attractive due to its importance in many vision surveillance problems. It is a challenging topic due to the inter/intra changes, occlusion, and pose variations involved. In this paper, we propose a pyramid-attentive framework that relies on multi-part features and multiple attention to aggregate features of multi-levels and learns attention-based representations of persons through various aspects. Self-attention is used to strengthen the most discriminative features in the spatial and channel domains and hence capture robust global information. We propose the use of part-relation attention between different multi-granularities of features' representation to focus on learning appropriate local features. Temporal attention is used to aggregate temporal features. We integrate the most robust features in the global and multi-level views to build an effective convolution neural network (CNN) model. The proposed model outperforms the previous state-of-the art models on three datasets. Notably, using the proposed model enables the achievement of 98.9% (a relative improvement of 2.7% on the GRL) top1 accuracy and 99.3% mAP on the PRID2011, and 92.8% (a relative improvement of 2.4% relative to GRL) top1 accuracy on iLIDS-vid. We also explore the generalization ability of our model on a cross dataset.

**Keywords:** computer vision; deep learning; person re-identification; attention; temporal aggregation; multi-granularities

## 1. Introduction

Person re-identification is the process of retrieving the best matching person in a video sequence across the views of multiple overlapping cameras. It is an essential step in many important applications such as surveillance systems, object tracking, and activity analysis. Re-identifying a person involves simply assigning a unique identifier to pedestrians captured within multiple camera settings. However, it is very challenging due to occlusion, large intra-class and small inter-class variation, pose variation, viewpoint change, etc.

Person re-identification is conducted on images [1–7] or videos [8–14]. Early approaches to this process can be classified into two main branches: either to develop handcrafted features or to develop machine learning solutions for optimizing parameters. Recently, the use of deep learning feature extraction has become very popular due to its success in many vision problems.

Nowadays, videos have gained more attention because they make use of the benefits of rich temporal information rather than using static images which suffer from limited content. Several related works have extracted frame features and aggregated them into maximum or average pooling [11,15], while other studies using temporal information have used temporal aggregates such as long- and short-term memory (LSTM) or recurrent neural networks (RNNs) [8,9,16,17].

Lately, "attention" has attracted more interest due to its strength in retaining important features and removing irrelevant ones. Several works have been designed using spatial-temporal attention [12,15] or recurrent attention [9,18,19], or have developed functions to re-weight frames [13,20] or use 3D convolution neural networks (CNNs), such as in [21].

Several studies have focused on developing part-based models, combined with global features [1,4,5,7,15]. Some works have also used an auxiliary dataset [22] to define an attribute [1,7] or have used pose estimation [6,15], while others have extracted features on a multi-scale [2,3].

Self-attention was introduced by Ashish et al. [23] in 2017. It complements CNNs and helps systems with multi-level dependencies to focus on important features. It has successfully achieved better performance in many linguistic tasks such as summarization, translation, reading comprehension, and sentence representation.

In this study, inspired by research on attention and a combination of studies on global and local features extraction, we have developed a multi-part feature extractor with a multi-attention framework. As such, the contributions of our study can be summarized as follows:

- Developing multilevel local features representation to overcome missing parts and misalignment in global representation. The integration between multi-local partitions makes the representation more generalizable, as the person is viewed with multiple granularities.
- Proposing a novel way to use self-attention in an image using a block of convolution layers to capture the most generalized information.
- Using channel attention in addition to spatial attention to capture correlation in all directions. This uses the relation between channels instead of only a spatial aspect.
- Introducing self-part attention between the multi-levels of features to benefit from the relationships between the parts in multi-granularities and produce better representations of each part.

The rest of the paper is organized as follows: The related works are discussed in Section 2; the proposed approach in Section 3; the experimental results in Section 4; and the conclusion and our intended future studies in Section 5.

## 2. Related Works

Person re-identification focuses on developing a discriminative feature set to represent a person. Early studies in this area have developed handcrafted features such as the local binary pattern (LBP) histogram [24], histogram of gradient (HOG) [25], and local-maximal-occurrence (LOMO) [26]. Other studies use a combination of different features [27,28].

After that, various deep learning models have been presented and show better performance compared to handcrafted features. For example, Shangxuan et al. [29] combined handcrafted features (ensemble of local features—ELF) with features extracted from CNNs, thereby making the CNN features more robust.

Video-based re-identification research focuses on three topics: feature extraction, temporal aggregation, and the attention function. Feature extraction refers to selecting the best features—global, part-based, or a combination of both—that represent a person. Temporal aggregation is the method by which each frame's features are aggregated to construct video-sequence features. The attention function is used to learn how important features can be strengthened and irrelevant ones suppressed.

Features are extracted from each frame and aggregated using the temporal average pooling, as provided in [30]. An RNN is designed to extract features from each frame and capture information across all time steps to get final feature representation [11]. Two stream networks, one for motion and the other for appearance, are used together for aggregating with the RNN [14]. Handcrafted features such as color, texture, and LBP with LSTM are used to prove the importance of LSTM in capturing temporal information. LSTM is used to build a timestamped frame-wise sequence of pedestrian representation that allows discriminative features to be accumulated to the deepest node and prevents

non-discriminative ones from reaching it [9]. Thus, two CONV-LSTM are proposed: one for getting spatial information and the other for capturing temporal information [18].

The global features are combined with six local attributes learned from the predefined RAP dataset [22]. Then, the frames are aggregated by re-weighting each frame using the attribute confidence function [31]. Co-segmentation [32] detects salient features across frames by calculating the correlation between multiple frames of the same tracklet and aggregating the frames using temporal attention. Multi-level spatial pyramid pooling is used to determine an important region in the spatial dimension and uses RNNs to capture temporal information [16]. A tracklet is divided into multiple snippets and learns co-attention between them [12]. The self and collaborative attention network (SCAN) [10] uses the self-attention sub-network (SAN) to select information from frames in the same video, and the collaborative attention sub-network (CAN) obtains the across-camera features. TAM is designed for generating weights that represent the importance of each frame, and the spatial recurrent model (SRM) is responsible for capturing spatial information and aggregating frames by RNN [13].

Spatial-Temporal Attention-Aware Learning (STAL) [15] presents a model that extracts global and local features, which is trained on the MPII human pose dataset [33] for learning body joints and attention branch for learning STAL. The relation module (RM) is designed to determine the relation between each spatial position and all other positions. It obtains the temporal relation among all frames according to the defined relational guided spatial attention (RGSA) and relational guided spatial attention (RGTR) [34]. The multi-hypergraph can be defined as multi-nodes where each node is in a different spatial granularity [35]. Attentive feature aggregation is presented to explore features along channel dimensions in different granularities and divides into groups. Each group has a different granularity than the attentive aggregate used [36]. Finally, intra/inter frame attention is proposed for re-weighting each frame [20].

Unlike the use of spatial and temporal attention in the previous methods, we propose the use of multiple attention for mining tensor vectors. Thus, we use the correlation from multiple directions and the advantages of channel dimension and part interaction to select important features from various dimensions. Taking advantage of multi-scale approaches, we propose a multi-part pyramid for exploring a person from multiple views that aims to extract discriminative and robust features.

## 3. Our Approach

Our model aims to explore pedestrians using the pyramid multi-part features with multi-attention (PMP-MA) for video re-identification. Pyramid multi-part features (PMP) detects pedestrians with multiple granularities to capture different details. Multiple Attention (MA) is applied to obtain the most important features in multiple domains. The combined approach gets the most robust features in multi-granularity levels and multiple domains. We start with a discussion of the overall architecture, followed by the details of the different model parts.

### 3.1. Overall Architecture

Initially, a group of pedestrian videos is fed into a backbone network X = {xt}1:T, where T is the number of frames. The backbone network is based on ResNet50. The output of the backbone is split into three independent branches; each branch represents a different granular level, as shown in Figure 1. Then, we apply self-spatial and channel attention to improve the feature extraction and obtain the best spatial channel and part attention to obtain the relation between parts in multiple granularities. Subsequently, the outcomes of the three branches are concatenated to learn the most robust features for representing a person at multiple levels. After that, we apply temporal attention pooling (TAP) [30] to aggregate the frame-level features and temporal information into a video-sequence feature representation. Finally, we use a classification layer with an intermediate bottleneck layer

to generalize the representation among the set of tracklets used in training before loss function. Two loss functions are used: triple loss and cross-entropy.

**Figure 1.** Overall Architecture of the System.

### 3.2. Pyramid Feature Extractor

To enhance the representation of person-reidentification, this model uses a pyramid feature extractor to represent a pedestrian with multi-granularities level for each frame. It represents the pedestrian by three granularities: global representation $gl_t$, coarse-grained part representation $l_k$ and fine-grained part representation $l_{2k}$. The feature map of global representation is $gl = \left\{ gl_t \middle| gl_t \in R^{h*w*c} \right\}_{t=1:T}$ where $h$, $w$, and $c$ represent the height, width, and channel size of the feature map. The features of the multi-part branches are partitioned into *K and 2\*K* horizontal parts $l_k = \left\{ l_{kt} \middle| l_{kt} \in R^{k*c} \right\}_{t=1:T}$ $l_{2k} = \left\{ l_{2kt} \middle| l_{2kt} \in R^{2k*c} \right\}_{t=1:T}$ where k represents the number of horizontal parts. Initially, the backbone network extracts an initial feature map. Then, it splits into three branches and extracts three different feature maps to represent the same person through separate convolution layers (based on Resnet50). Besides that, it extracts complementary features to use as attention features.

The first branch is the global extractor that captures the global features of each pedestrian. It represents the whole frame with one vector $gl$ and produces attention vector $p_{att1} = \left\{ p_{att1t} \middle| p_{att1t} \in R^{K*C} \right\}$ for part attention. The second branch divides the feature map into *K* parts to capture the local features for each part, captures the coarse-grained parts of each pedestrian $l_k$, and represents the frame with K vectors. Beside that, it produces attention vector $g_{att1} = \left\{ g_{att1t} \middle| g_{att1t} \in R^{h*W*C} \right\}$ for spatial and channel attention. The third branch divides the map into 2\**K* parts. It captures the fine parts of each pedestrian and represents a frame with 2\**K* vectors $l_{2k}$ and produces the attention vectors $g_{att2} = \left\{ g_{att2t} \middle| g_{att2t} \in R^{h*W*C} \right\}$ and $p_{att2} = \left\{ p_{att2t} \middle| p_{att2t} \in R^{2K*C} \right\}$. The model extracts global, coarse parts and fine parts features from the three branches and fuses them to obtain the most robust features.

### 3.3. Self-Spatial Channel Attention

Self-attention is an intra-attention mechanism to re-weight feature *F* with an attention score matrix *S(x,y)* for strengthening the discriminative feature. It is designed to learn the correlation between one pixel and all other positions. It can explicitly capture global

interdependencies. There are different techniques for calculating the attention score; in this work, the attention score $S$ is calculated by computing the relation and interdependencies of two attention branches using the attention dot-product [21]. We compute the tensor-matrix-multiplication of $x$ and $y$, where $x$ and $y$ are two feature attention vectors, and apply the *softmax* function to get $S$ (Equation (1)), as shown in Figure 2. Self-attention is used as a residual block [37], which sums the output of $S * F$ to the original $F$. Equation (2) expresses the output of the self-attention block $f\_att$. where x, y, F$\rightarrow$ feature vectors.

$$S(x,y) = softmax\left(x * y^T\right) \tag{1}$$

$$F = f\_att(F,S) = (F + S(x,y) * F) \tag{2}$$

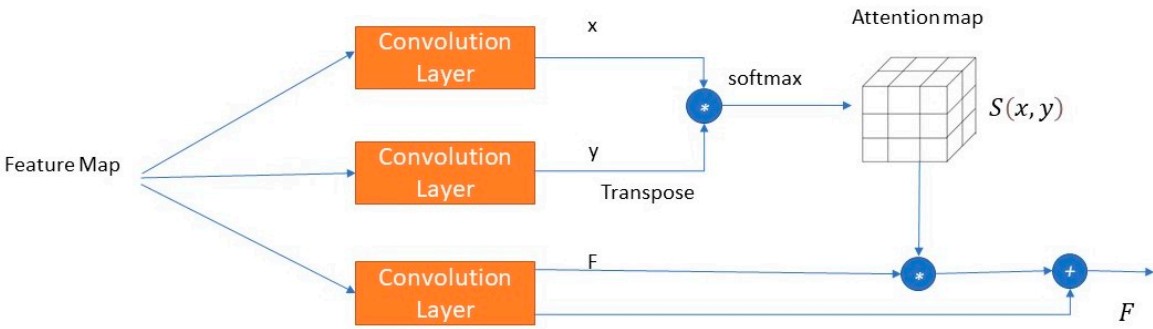

**Figure 2.** Overall Architecture of the Self-attention.

Using three branches satisfies the need of self-attention. Spatial-channel self-attention uses $g_{att1}$ and $g_{att2}$ for building an attention map that represents the relation between spatial and channel. Spatial attention enriches the spatial relationship. Let $g_{att1}$ and $g_{att2}$ be attention branches; when rearranged ($g_{att1}$, $g_{att2} \in R^{hw*c}$, $R^{hw*c}$), each spatial position is described by the $c$ channel. Hence, the spatial attention score after applying function $S$ is $s1 \in R^{hw*hw}$ (Equation (3)), which represents the relation between each position and all positions. Then, we apply the $f\_att$ function for $gl$ (Equation (4)).

$$s1 = S(g_{att1}, g_{att2}) \tag{3}$$

$$glsp = f_{att}(gl, s1) \tag{4}$$

At this step, we construct global self-channel attention. This type of attention aims to strengthen the weight of important channels and suppress less important ones. First, we apply a summary layer over the three branches ($g_{att1}$ $g_{att1}$ *and* $glsp$) to sum all spatial position and focus on the channel feature for each branch. For the global attention branches, we calculate $g_{avg1}$, $g_{avg2}$ *and* $glsp_{avg} \in R^{c*1}$ (Equations (5)–(7)) and then compute the channel attention score by applying function $S$ for $g_{avg1}$, $g_{avg2}$ (Equation (8)); the result is the channel score ($s2 \in R^{c*c}$). The $s2$ score represents the relation between channels. Finally, we apply the $f\_att$ function over channel score $s2$ and the third branch $glsp_{avg}$, and compute the self-spatial with channel attention $glsch$ (Equation (9)). The flow of the whole process is shown in Figure 3, where N is the number of pedestrians in each batch and t is the number of frames describing each pedestrian.

$$g_{avg1} = \frac{\sum_{h,w} g_{att1}}{h * w}, \tag{5}$$

$$g_{avg2} = \sum_{h,w} g_{att2} / h * w \tag{6}$$

$$glsp_{avg} = \sum_{h,w} glsp / h * w \tag{7}$$

$$s2 = S(g_{avg1}, g_{avg2}) \tag{8}$$

$$glsch = f\_att(glsp_{avg}, s2) \tag{9}$$

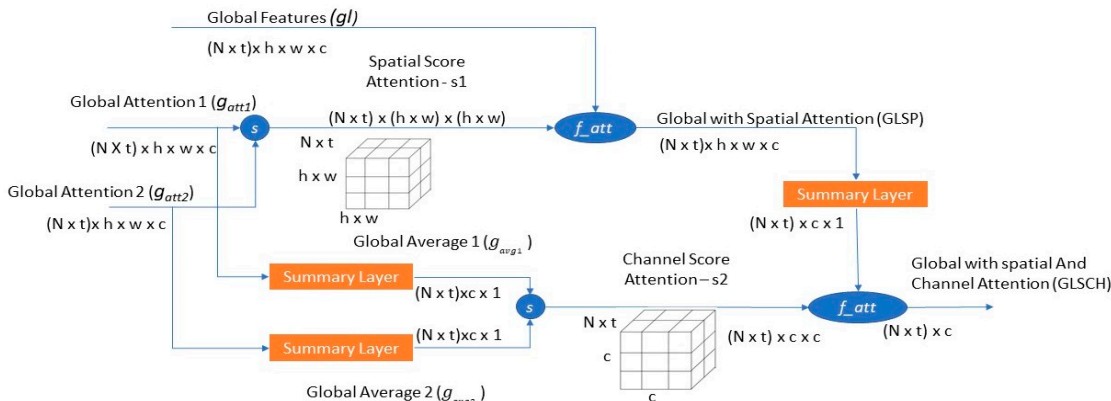

**Figure 3.** Self-Spatial with Channel Attention.

### 3.4. Self-Part Attention

The local feature extractor is better at learning specific parts than the global one. Pedestrians can be partitioned into several parts, and two local branches are used to see a pedestrian with coarse-grained and fine-grained parts. Part attention is then used to learn the robust features of each coarse part by computing the relationship among the parts in different branches. It applies the $S$ function over the attention vector from the global branch ($p_{att1}$) and attention from fine parts ($p_{att2}$). The $s3$ score represents the relation between the two granularity parts. Then, the coarse part with attention ($lKpA$) is computed by applying $f_{att}$ over $s$ and the $K$ features. After that, it rearranges $lKpA$ vector to let each part be represented by (N × t) × c. This results in enriching each part with discriminative features for each $k$ part, as shown in Figure 4.

$$s3 = S(p_{att1}, p_{att2}) \tag{10}$$

$$lKpA = f_{att}(l_k, s3) \tag{11}$$

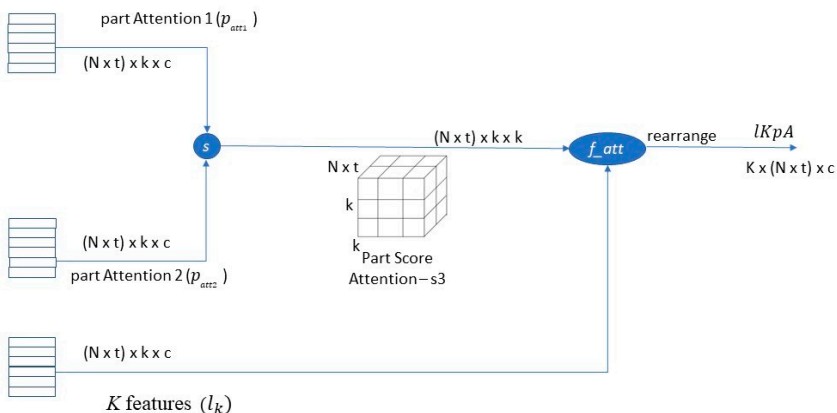

**Figure 4.** Part Attention.

### 3.5. Temporal Attention

After extraction frame level features, temporal attention pooling (TAP) [30] applies to generate video level representation. The global features after spatial-channel attention ($glsch$), each part in coarse-grained after part attention ($lKpA_p$), and each part in fine-grained part representation ($l_{2kp}$) are fed into TAP. Initially, the convolution layer is used to

reduce the features from $c$ to $c'$; thus, it is more generalized. Then, the other convolution layers with the *softmax* function are used to calculate the attention scores as shown in Figure 5. Finally, the aggregation layer directly takes each pedestrian described by t frames and aggregates the frames into a single generalized feature vector.

$$gl_v = TAP(glsch) \tag{12}$$

$$l_{vp1} = TAP(LKpA_p) \tag{13}$$

$$l_{vp2} = TAP(l_{2kp}) \tag{14}$$

where $gl_v$ is a feature vector of global video representation, $LKpA_p$ is a feature vector of part $p$ in coarse-grained frame representation, $l_{2kp}$ is a feature vector of part $p$ in fine-grained frame representation, $l_{vp1}$ is a feature vector of part $p$ in coarse-grained video representation, $l_{vp2}$ is a feature vector of part p in fine-grained video representation.

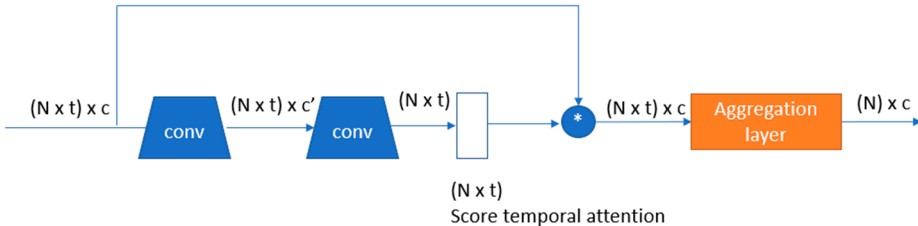

**Figure 5.** Temporal Attention.

*3.6. Objective Function*

Our objective function $L$ is a combination of the hard triplet loss $L_{tri}$ and cross-entropy loss $L_{ce}$ in both the global features and each local part, as shown in Equation (11).

$$L = L_{tri}(gl_v) + L_{ce}(gl_v) + \sum_k (L_{tri}(l_{vp1}) + L_{ce}(l_{vp1})) + \sum_{2k} (L_{tri}(l_{vp2}) + L_{ce}(l_{vp2})) \tag{15}$$

Further details of the functions used are as follows:

- Triplet Loss ($L_{tri}$): The distance between pairs from the same pedestrian is minimized (reduced intra-class), where the distance between pairs of different pedestrians is maximized (increased inter-class). We use the hard triplet loss that selects the hardest example for the positive and negative pairs, where $f_A, f_+, f_-$ are the anchor, positive features, and negative features, respectively.

$$L_{tri} = \frac{1}{N} \sum_i^N m + \max_N (D(f_A, f_+)) - \min_N (D(f_A, f_-)) \tag{16}$$

- Cross entropy ($L_{ce}$): It is used to calculate the classification error between pedestrians, where $N$ is the number of pedestrians. Where, $p_i$ and $q_i$ are the identity and prediction of sample $i$.

$$L_{ce} = \frac{1}{N} \sum_{i=1}^N p_i \log q_i \tag{17}$$

For the classification layer in the cross-entropy function, we used the intermediate bottleneck to reduce dimension and make it more generalized.

**4. Experimental and Discussion**

*4.1. Datasets Used*

The proposed model was evaluated on four widely used video-based person re-identification datasets, i.e., iLIDs-VID [38], PRID2011 [39], DukeMTMC-VideoReID [40], and motion analysis and re-identification (MARS) [41]. The standard evaluation metric for the model is the mean average precision score (mAP) and the cumulative matching

curve (CMC) at Rank1, Rank5, Rank10, and Rank20. CMC measures the matching accuracy of a person, while mAP measures the accuracy of the result performance. Samples of the dataset used are shown in Figure 6.

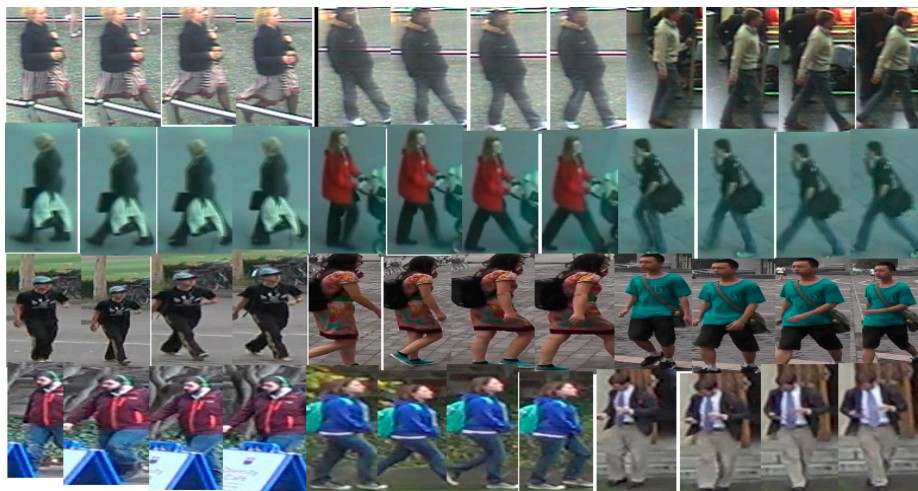

**Figure 6.** A Sequence of Person Tracklets from Different Datasets.

### 4.1.1. iLIDs-VID

The iLIDS-VID dataset [38] contains 600 tracklets of 300 persons captured from two non-overlapping cameras. Each tracklet has a length of 23–193 frames. A total of 150 pedestrians were used for training and 150 for testing This dataset was taken in a crowded airport arrival hall.

### 4.1.2. PRID2011

The PRID2011 dataset [39] contains 400 tracklets of 200 persons captured from two non-overlapping cameras. A total of 385 pedestrians were under camera A and 749 under camera B. Each tracklet has a length of 5–675 frames. Following a previously published protocol [38], we selected tracklets with more than 27 frames; 89 pedestrians were selected for training and 89 for testing. This dataset was taken in an uncrowded outdoor area with varying degrees of illumination and viewpoints. In iLIDs-VID and PRID2011, we randomly split them into train/test pedestrians. This is repeated 10 times for computing averaged accuracies performance.

### 4.1.3. DukeMTMC-VideoReID

The DukeMTMC-VideoReID dataset [40] is one of the largest datasets in video-based person re-identification. It contains 2196 tracklets of 702 persons captured for training and 702 tracklets of 702 persons in query, while the gallery contains 2636 tracklets for 1110 persons. Each tracklet has a length of 1–9324 frames. Eight cameras were used to capture this dataset, which was taken in a crowded outdoor area with varying degrees of illumination and occlusion, viewpoints, and backgrounds.

### 4.1.4. MARS

The MARS dataset [41] is another large dataset in video-based person re-identification. It contains 8298 tracklets of 626 persons captured for training and 1980 tracklets of 626 persons in query, while the gallery contains 9330 tracklets of 626 persons. Each person was captured with at least two cameras. Each tracklet has a length of 2–920 frames. This sequence was extracted using the DPM pedestrian detector [42] and GMMCP tracker [43]. The videos were taken in a crowded outdoor space with six cameras and have varying viewpoints and complicated occlusion. We used post-processing re-ranking [44] as a post-processing step

to enhance the results of the test phase due to multiple appearances by the person in the gallery rather than only one as in other datasets; a similar approach is used in [21,32].

### 4.2. Implementation Details

We compared the most efficient deep learning frameworks and selected Pytorch [45] to implement our model. Pytorch's consequent performance is enhanced significantly; moreover, it has off-the-shelf person re-identification libraries. The images were resized to $216 \times 423$ and randomly cropped to $192 \times 384$. Then, they were normalized using the RGB mean and standard deviation. We used the ResNet50 pre-trained on ImageNet for extracting features per frame. Our backbone uses the first two layers and part one of layer three of the ResNet50. The second part of layer three of the ResNet50 is duplicated in our three independent branches. We removed the last layer to increase the resolution of the final feature map so more details could be preserved—which is beneficial for further multilevel learning—and replaced it with our PMP model. We used K = 6 and $2 \times K = 12$ for multi-local partitioning; the effects of this choice are discussed in Section 4.6. In training to form the N batch, first, Pid pedestrians and Seq different sequences for each Pid were selected randomly, and each Seq sequence had T frames. The total sequence is Pid* seq in each batch. We used T = 4 and Seq = 2 in small datasets (iLIDs-VID and PRID2011) and T = 4 and Seq = 4 in large-scale datasets (DukeMTMC-VideoReID and MARS); these values are further discussed in Section 4.3. The network was trained using the Adam optimizer [46] with the following hyper-parameters: the initial learning rate = 0.0003, weight decay = $5 \times 10^{-4}$, gamma = 0.1, h = 24, w = 12, c = 1024, and c' = 256.

The implementation was processed over two machines with different GPU specs (one for small datasets and one for large datasets). Table 1 shows the training time of each dataset:

**Table 1.** Summary of implementation training time.

| # | Dataset | Machine Spec | Training Time |
|---|---------|--------------|---------------|
| 1 | iLIDS-VID | NVIDIA GeForce GTX 1070 8GB DDR5 | 6 h |
| 2 | PRID2011 | | 4 h |
| 3 | DukeMTMC-VideoReID | NVIDIA TESLA T4 32GB DDR6 | 30 h |
| 4 | MARS | | 36 h |

### 4.3. Comparison with State-of-the-Art

In this section, we compare our model with other state-of-the-art models, including Global Guided Reciprocal learning(GRL) [47], BICnet [48], Clip similarity [21], Rethink temporal fusion [20], TACAN [12], MG-RAFA [35], MGH [36], SCAN [10], Co_segment [32], STAL [15], STN [19], and Attribute disentangling [31]. Table 2 shows the recent video-based person re-identification models along with a summary of their technique and year of publication.

Table 3 shows the performance of our PMP-MA model versus the most recent video-based person re-identification models when applied to the small datasets. Our model improves Rank1 relative to the best R1 so far (GRL) by 2.7% on PRID2011(See multiple moderate and hard examples for PRID2011 in Tables S1 and S2 in supplementary material file) and relative to the best R1 (GRL) by 2.4% on iLIDS-vid (See multiple moderate and hard examples for iLIDS-vid in Tables S3 and S4 in supplementary material file) due to the improvement in multi-level learning and by using multi-attention instead of one.

**Table 2.** Summary of Recent Video-Based Person Re-Identification Techniques.

| Model | Technique | Year |
|---|---|---|
| GRL | • Correlation maps between frames<br>• Temporal learning | 2021 |
| BICnet | • Spatial representation across frames<br>• Temporal relations | 2021 |
| Clip similarity | • 3D<br>• Reranking | 2020 |
| Rethink temporal fusion | • ResNet50<br>• Intra/inter frame attention | 2020 |
| TACAN | • Clip attention | 2020 |
| MG-RAFA | • ResNet50<br>• Multigranularity<br>• Spatial attention | 2020 |
| MGH | • ResNet50 + multi<br>• Granularity + hypergraph | 2020 |
| Scan | • ResNet 50<br>• Att between frames + collaboration att across cameras | 2019 |
| Co-segmentation | • SE-ResNet50 + co-segmentation-based spatial-temporal attention | 2019 |
| STAL | • Global + local features<br>• Spatial-temporal attention | 2019 |
| STN | • GoogleNet + BiLSTM | 2018 |
| Attribute disentangling | • Resnet 50 + N local attribute group + TAM | 2019 |

**Table 3.** Comparing the Models' Performance Using Small-Scale Datasets.

| Dataset<br><br>Model | iLIDS-vid | | | | | PRID2011 | | | | |
|---|---|---|---|---|---|---|---|---|---|---|
| | mAP | R1 | R5 | R10 | R20 | mAP | R1 | R5 | R10 | R20 |
| GRL | - | <u>90.4</u> | <u>98.3</u> | - | <u>99.8</u> | - | <u>96.2</u> | 98.3 | - | <u>99.8</u> |
| Clip similarity | - | - | - | - | - | - | 82.9 | 95.8 | - | 99.1 |
| Rethink temporalfusion | - | 87.7 | - | - | - | - | 95.8 | - | - | - |
| TACAN | <u>93.0</u> | 88.9 | - | - | - | 96.7 | 95.3 | - | - | - |
| MG-RAFA | - | 88.6 | 98 | - | 99.7 | - | 95.9 | <u>99.7</u> | - | **100** |
| MGH | - | 85.6 | 97.1 | - | 99.5 | - | 94.8 | 99.3 | - | **100** |
| Scan | 89.9 | 88.0 | 96.7 | - | - | 95.8 | 95.3 | 99.3 | - | - |
| Co-segmentation | - | 75.9 | 94.1 | - | - | - | - | - | - | - |
| STAL | - | 82.8 | 95.3 | <u>97.7</u> | 98.8 | - | 92.7 | 98.8 | 99.5 | **100** |
| STN | - | 57.7 | 81.7 | - | 94.1 | - | 87.8 | 97.4 | - | 99.3 |
| Attribute disentangling | - | 86.3 | 97.4 | - | 99.7 | - | 93.9 | 99.5 | - | **100** |
| **Our Model** | **95.3** | **92.8** | **99.3** | **100** | **100** | **99.3** | **98.9** | **100** | **100** | **100** |

All the values are in percentages. The values in bold are the best results and the underlined values are the second-best results.

Table 4 presents a comparison of the performances when applied to large-scale datasets. Our model improves Rank1 by 0.9% relative to the best R1 (BICnet) on DukeMTMC-VideoReID and improves mAP by 2.1% relative to the best MAP (BICnet) on MARS after reranking.

**Table 4.** Comparing the Models' Performances Using Large-Scale Datasets.

| Dataset / Paper | DukeMTMC-VideoReID | | | | | MARS | | | | |
|---|---|---|---|---|---|---|---|---|---|---|
| | mAP | R1 | R5 | R10 | R20 | mAP | R1 | R5 | R10 | R20 |
| GRL | - | - | - | - | - | 84.8 | **91.0** | <u>96.7</u> | - | 98.4 |
| **BICnet** | <u>96.1</u> | <u>96.3</u> | - | - | - | <u>86.0</u> | 90.2 | - | - | - |
| Rethink temporal fusion | - | - | - | - | - | 85.2 | 87.1 | - | - | - |
| Clip similarity | 88.5 | 89.3 | <u>98.3</u> | - | 99.4 | 83.3 | 83.4 | 93.4 | - | 97.4 |
| TACAN | 95.4 | 96.2 | **99.4** | **99.6** | - | 84 | 89.1 | 96.1 | - | 98.0 |
| MG-RAFA | - | - | - | - | - | 85.9 | 88.8 | **97** | - | 98.5 |
| MGH | - | - | - | - | - | 85.8 | 90.0 | <u>96.7</u> | - | 98.5 |
| Scan | - | 88.0 | - | - | - | 77.2 | 87.8 | 95.2 | - | 98.1 |
| Co-segmentation | 94.0 | 94.4 | 99.1 | - | - | 77.2 | 83.7 | 94.1 | - | - |
| Co-segmentation + re-ranking | 94.1 | 95.4 | <u>99.3</u> | - | 99.8 | 87.4 | 86.9 | 95.5 | - | 98 |
| STAL | - | - | - | - | - | 73.5 | 82.2 | 92.8 | <u>98</u> | - |
| STN | - | - | - | - | - | 69.1 | 80.5 | 91.8 | - | 96 |
| Attribute disentangling | - | - | - | - | - | 78.2 | 87.0 | 95.4 | - | <u>98.7</u> |
| **Our Model** | 96.3 | 97.2 | <u>99.3</u> | **99.6** | 100 | 78.3 | 86.2 | 96.3 | **98.3** | **98.9** |
| **Our Model+ re-ranking** | **96.3** | **97.2** | <u>99.3</u> | **99.6** | 100 | **88.1** | <u>90.6</u> | 96.6 | 97.9 | 98.6 |

All the values are in percentages. The values in bold are the best results and the underlined values are the second-best results.

### 4.4. Effectiveness of Increasing the Batch Size

Compared to small datasets (two cameras), large-scale datasets are more diverse, as they capture people from different views using many cameras (six–eight cameras). With large datasets, we found that increasing the batch size and the number of instances enhanced the accuracy of catching large variations in different poses and changes in view, as shown in Table 5. Rank1 and mAP are improved by 3.2% and 2.5%, respectively, when using a batch size of 32 and four instances, relative to using a batch size of eight and two instances and Rank1 on DukeMTMC-VideoReID. However, Rank1 and mAP are improved by 7.6% and 6.1%, respectively, when using a batch size of 32 and four instances relative to using a batch size of eight and two instances and Rank1 on MARS. Increasing the batch size would lead to more computation; thus, we cannot increase the batch size to more than 32 due to our limited capabilities. However, Rank1 and mAP are almost saturated on DukeMTMC-VideoReID, so much so that accuracyimproves by 2.3% in Rank1 when increasing the batch size from 8 to 16, but it increases by only 0.9% when increasing the batch size from 16 to 32. Unfortunately, we thought that MARS could be improved more if we increased the batch size by more than 32, as we found Rank1 to increase by 2.3% when increasing the batch size from 8 to 16, whereas Rank1 increased by 3.9% when increasing the batch size from 16 to 32.

**Table 5.** Comparing Our Model's Accuracy Using Different Batch Sizes and Instances.

| Dataset Batch Size | Instance | DukeMTMC-VideoReID | | | | | MARS | | | | |
|---|---|---|---|---|---|---|---|---|---|---|---|
| | | mAP | R1 | R5 | R10 | R20 | mAP | R1 | R5 | R10 | R20 |
| 8 | 2 | 93.8 | 94 | 99.1 | 99.6 | 99.6 | 82 | 83 | 87.5 | 92.7 | 95.0 |
| 8 | 4 | 92.6 | 92.1 | 98 | 99 | 99 | 80 | 80 | 82.5 | 88.7 | 92.0 |
| 16 | 2 | 95.3 | 95.8 | 99.3 | 99.6 | 100 | 83.8 | 84.6 | 89.6 | 94.6 | 96.1 |
| 16 | 4 | 95.4 | 96.3 | 99.3 | 99.3 | 100 | 84.2 | 84.9 | 90 | 95.6 | 96.7 |
| 32 | 2 | 96.1 | 97.2 | 99.3 | 99.3 | 100 | 87 | 89 | 96 | 97.1 | 98.3 |
| 32 | 4 | **96.3** | **97.2** | **99.3** | **99.6** | **100** | **88.1** | **90.6** | **96.6** | **97.9** | **98.6** |

All the values are in percentages.

Conversely, in the small dataset shown in Table 6, increasing the batch size and number of instances did not result in a similar improvement, and the variance of the results is almost saturated. For example, in the case of mAP, increasing the batch size from 8 to 32 and the number of instances from 2 to 4 produces only 0.5% extra accuracy. This is because it already has two cameras with a single tracklet for each person. Thus, with two instances, we already capture all the variations.

**Table 6.** Comparing Our Model's Accuracy Using Different Batch Sizes and Instances.

| Dataset | | iLIDS-vid | | | | | PRID2011 | | | | |
|---|---|---|---|---|---|---|---|---|---|---|---|
| Batch Size | Instance | mAP | R1 | R5 | R10 | R20 | mAP | R1 | R5 | R10 | R20 |
| 8 | 2 | 94.8 | 92 | 99.3 | 100 | 100 | 98.9 | 98.4 | 100.0 | 100.0 | 100.0 |
| 8 | 4 | 94 | 91.8 | 98 | 99 | 99 | 98.2 | 98.1 | 99.3 | 99.3 | 99.3 |
| 16 | 2 | 94.7 | 92.1 | 99.3 | 100 | 100 | 98.9 | 98.6 | 100.0 | 100.0 | 100.0 |
| 16 | 4 | 94.8 | 92.3 | 99.3 | 100 | 100 | 98.9 | 98.6 | 100.0 | 100.0 | 100.0 |
| 32 | 2 | 95.2 | 92.6 | 99.3 | 100 | 100 | 98.9 | 98.6 | 100.0 | 100.0 | 100.0 |
| 32 | 4 | 95.3 | 92.8 | 99.3 | 100 | 100 | 99.3 | 98.9 | 100 | 100 | 100 |

All the values are in percentages.

### 4.5. Cross-Dataset Generalization

Using a cross-dataset is a better way of measuring our model's generalization. It evaluates the ability of a system to perform on a different dataset than the training dataset. Each dataset is collected in different visual conditions and using different viewpoints. Often, models trained in one dataset perform badly on others. To evaluate our model under a more general setting, we used the iLIDS-VID dataset for training and PRID2011 for testing and compared our results to the STN [19] and RCN [11] models used in the same setting.

The results in Table 7 show that the proposed model is more generalizable than the current best state-of-the-art models. Our model improves Rank1 by 11.5%. However, an accuracy of 40% is expected due to the challenges of using a cross-dataset. Slight improvements have been verified for Rank5, Rank10, and Rank20. The use of a cross-dataset is an open research issue. Our model certainly achieves better performance, but more optimization is required to enhance the generalization performance.

**Table 7.** Model Accuracy Using a Cross-Dataset.

| Model | Rank1 | Rank5 | Rank10 | Rank20 |
|---|---|---|---|---|
| RCN | 28 | 57 | 69 | 81 |
| STN | 29.5 | 59.4 | - | 82.2 |
| Our Model | **40** | **63** | **70.2** | **85** |

All the values are in percentages.

### 4.6. Ablation Study

In this section, we show the contribution of each component of the proposed model. Table 8 summarizes the performance of each module separately when applied to iLIDS-VID. First, we evaluated Resnet50 optimized with triple loss and cross entropy without any add-on component as pretraining model. Then, it was evaluated with TAP [30]. We discovered that TAP aggregation improved mAP by 5.9% and Rank1 by 5%. After that, the baseline was evaluated, where the baseline is the pretraining model with TAP aggregation in addition to bottleneck layer in the classification layer before the cross-entropy loss function and removed layer 4 of ResNet50. Experiments prove that adding an intermediate bottleneck in the classification layer (cross-entropy function), improved the baseline mAP by 9.4% and Rank1 by 9%. The reason for this was that the bottleneck layer compressed features representation to find the best fit and become more generalized.

**Table 8.** The Effects of the Components of the PMP-MA Model on iLIDS-VID.

| Model Component | mAP | Rank1 | Rank5 | Rank10 | Rank20 |
|---|---|---|---|---|---|
| Pretrain | 75.3 | 68.3 | 86 | 96 | 96 |
| Pretrain with TAP | 81.2 | 73.3 | 89.3 | 96.0 | 97.3 |
| Baseline n | 84.7 | 77.3 | 94.3 | 98.0 | 99.3 |
| Baseline + PMP | 89 | 87 | 97 | 100 | 100 |
| Baseline + Self-spatial attention | 86.1 | 80.3 | 94.7 | 98.0 | 99.3 |
| Baseline + Self-spatial and channel attention | 88.3 | 82 | 96 | 99.3 | 100 |
| Baseline + PMP + Self-spatial and channel attention | **92.4** | **90.3** | **99.3** | **100** | **100** |
| PMP-MA | **95.3** | **92.8** | **99.3** | **100** | **100** |

All the values are in percentages.

Secondly, we tested each component of our system in the fine-tuning stage. First, our PMP was added to baseline. As shown in Table 8, the PMP component can achieve mAP of 89% and 87% in Rank1 and 100% in Rank 10, which improves the system relative to the baseline by 4.3%, 9.7%, and 4 %. The reason is that the PMP extracts pedestrians with multi-level partitions and fuses them. The multi-level partition overcomes the fine parts that are usually missed if we extract global features with one-level partitioning. Then, the self-part attention was tested by the replaced PMP component. The spatial attention component can achieve mAP of 86.1% and 80.3% in Rank1, which improved the system relative to the baseline by 1.4% and 3%. After that, we added channel attention along with spatial attention; it improved the system relative to the baseline with spatial with the mAP increasing by 2.2% and Rank1 by 2%.

We evaluated our PMP component, which was combined with the self-spatial and channel components. Our component can achieve mAP of 92.4%, Rank1 of 90.3%, and Rank5 of 99.3%, which improves the system relative to the baseline by 7.7%, 14%, and 5%. Finally, we tested the last component of our model by adding part attention to spatial and channel attention, which improved the system by increasing the mAP by 2.9% and Rank1 by 2.5%. It is obvious that our model achieves better mAP and Ranks accuracies, which show the effectiveness of the PMP-MA framework.

### 4.7. The Effect of Using Different Parts (K)

The number of parts (k) was evaluated during the monitoring of our model's performance. In particular, we analyzed k = 2, 3, 4, 6, 8, 10, and 12. The results are shown in Figure 7. We found k = 6 is the best.

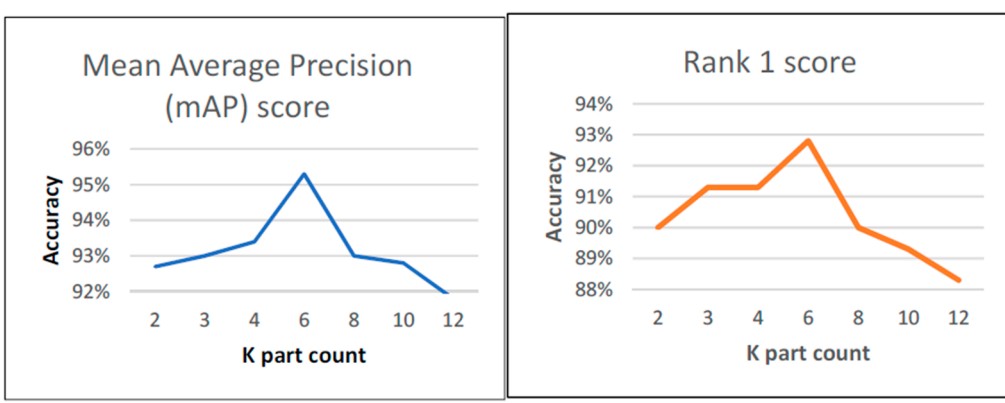

**Figure 7.** Analyzing the Effect of different K-Parts on the mAP and Rank1 score.

### 4.8. Analysis of Part Attention (K)

There are many ways to calculate score attention; in this work, we used self-part attention (Figure 4), comparing it against using one attention branch (Figure 8). Table 9

lists the effect of self-part attention on the coarse branch. Using self-part attention instead of the attention branch in component 3 improved mAP by 0.5% and Rank1 by 0.8% on the iLIDS-VID over the results of component 2 and by 0.6% and 1.1%, respectively, on the PRID dataset. In comparison to the results of component 1, which has no part attention, it improved mAP by 2.9% and Rank1 by 2.8% on the iLIDS-VID and by 2.3% and 4.5%, respectively, on the PRID dataset relative to not using part attention as in component 1.

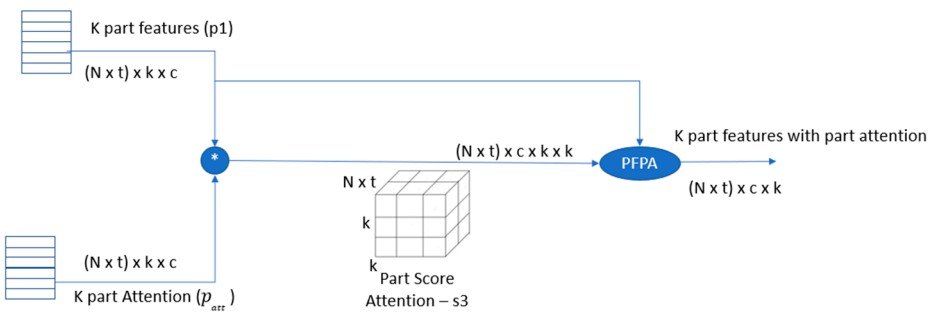

**Figure 8.** Part Attention with One Attention Branch.

**Table 9.** The Effect of Self-Part Attention on the Coarse Branch.

| | Dataset | iLIDS-VID | | | | PRID | | | |
|---|---|---|---|---|---|---|---|---|---|
| # | Model Component | mAP | Rank1 | Rank5 | Rank10 | mAP | Rank1 | Rank5 | Rank10 |
| 1 | Proposed model without part attention | 92.4 | 90.3 | 99.3 | 100 | 96.7 | 94.4 | 100 | 100 |
| 2 | Proposed system with part attention with one branch | 94.8 | 92 | 99.3 | 100 | 98.7 | 97.8 | 100.0 | 100.0 |
| 3 | Proposed system with self-part attention | 95.3 | 92.8 | 99.3 | 100 | 99.3 | 98.9 | 100 | 100 |
| 4 | Proposed system with two self-part attention | 93 | 91 | 99.3 | 100 | 97.0 | 94.4 | 100 | 100 |

Table 9 shows that using two-part attention in coarse and fine branches, as in component 4, worsens Rank1 by 1.8% and mAP by 2.3% on the iLIDS-VID relative to using only coarse part attention and by 4.5% and 2.3%, respectively, on PRID.

Thus, if we use part attention on the coarse branch, we focus on the important features on the coarse branch and extract fine details from the fine branch. Hence, not using any part attention in both branches does not capture discriminative part features as in component 1, whereas overusing two-part attention in both branches results in losing focus of important coarse part features as in component 4. It is obvious that using part attention in the coarse-grained branch achieved best accuracy due to focus on discriminative features as well as fine details.

## 5. Conclusions

This paper proposes a PMP-MA extractor for video-based re-identification. A multi-local model can learn generic and specific (multiple localization) features of each frame. To take full advantage of the pyramid model, we used self-spatial and channel attention, which enabled us to specify the quality of each spatial feature and channel to enrich the features vector. Multiple local parts were used to learn a specific part with two scales and part attention. The PMP system is complemented with TAP, which is used to extract temporal information among frames. The evaluation was done against four challenging datasets, where the proposed model achieved better performance in three datasets, that is, 2.4% over the iLIDs-VID, 2.7% over PRID2011, 0.9% over DukeMTMC-VideoReID, and 11.5% over the cross-dataset. The PMP-MA extractor is a well-designed extractor that can extract and

fuse robust features from multiple granularities. Potential applications of this approach in other computer vision problems include object tracking and image segmentation or video object segmentation. In the future, we plan to add positional encoding to get the model to pay more attention to important frames. Moreover, we will try to enhance our pyramid model to reduce the complexity of the system components to overcome the necessity of having a GPU with a betterVRAM capacity.

**Supplementary Materials:** The following are available online at https://www.mdpi.com/article/10.3390/bdcc6010020/s1, Table S1: PRID Dataset, simple-moderate examples; Table S2: PRID Dataset, Hard Example; Table S3: iLIDS-VID Dataset, Rank1 is the correct matching tracklet; Table S4: iLIDS-VID Dataset (Hard Example).

**Author Contributions:** Conceptualization, R.M.B. and E.E.H.; methodology E.E.H.; Project administration, M.B.F.; Software, R.M.B.; Writing—original draft, R.M.B. and E.E.H.; Writing—review & editing, M.E.R. and M.B.F. All authors have read and agreed to the published version of the manuscript.

**Funding:** This research received no external funding.

**Institutional Review Board Statement:** Not applicable.

**Informed Consent Statement:** All the datasets used in this study, i.e., iLIDS-VID and PRID2011 were published in public Internet for research purpose so no consents are obtained.

**Data Availability Statement:** The dataset sources are listed in the paper, which are cited as [38–41].

**Conflicts of Interest:** The authors declare no conflict of interest.

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
