# Peer review of "Person Re-Identification via Pyramid Multipart Features and Multi-Attention Framework"

_2504-2289, doi:10.3390/bdcc6010020_

Round 1

Reviewer 1 Report

The paper proposes a deep learning framework for person re-identification that relies on multi-part features and multiple attention to aggregate features captured in multi-levels. The basis of the framework is the self-attention mechanism that tries to strengthen the most representative features. The framework achieved 98.9% mAP on PRID2011 dataset and 92.8% Top1 Acc on iLids-vid dataset.  

The paper overall shows a very interesting approach that was tested in 4 different datasets and the results were compared against several other proposals.

My overall suggestions are:

  • The text should be reviewed and the authors should keep a uniformity of what should be in capital letters and what should not.
  • In Table 2, why there is "BICnet" if there is no value associated with it? 
  • The model proposed is very complex, would be great to see a table with the training time for each dataset. Moreover, if possible, try to compare this time against the other models evaluated. 

Reviewer 2 Report

The article is interesting and raises the interesting issue of re-identification of persons. It can be classified as a paper presenting the application of neural networks in practical solutions. The authors of the paper presented many different alternative solutions to the problem and showed that their solution is slightly better than the others.

In the conclusions, the purpose of the developed algorithm and the possibilities for its use should be clearly highlighted. It is also worth emphasising what the potential advantage of the solution over others will be. The percentages results do not fully represent the possible benefits of the approach.

Amendments to be made:

  • It would be useful to include and comment on an instance with several people in one image.
  • Language corrections are required
  •  

    In Figure 3 and 4, the description of the input features (N xt) x h x w x c is illegible.

  •  

    Figure 5 - the input signal to the aggregation layer is unclear.
  •  
